# Auto Encoding Knockoff generator for FDR Controlled Variable Selection

## Abstract

A new statistical procedure (Model-X Candès et al. (2018)) has provided a way to identify important factors using any supervised learning method controlling for FDR. This line of research has shown great potential to expand the horizon of machine learning methods beyond the task of prediction, to serve the broader needs in scientific researches for interpretable findings. However, the lack of a practical and flexible method to generate knockoffs remains the major obstacle for wide application of Model-X procedure. This paper fills in the gap by proposing a model-free knockoff generator which approximates the correlation structure between features through latent variable representation. We demonstrate our proposed method can achieve FDR control and better power than two existing methods in various simulated settings and a real data example for finding mutations associated with drug resistance in HIV-1 patients.

## 1 Introduction

In the past decades, the machine learning methods have achieved great advancement in improving prediction accuracy. However, prediction accuracy is not the sole quest of 'big data' analysis. A universal aim across a lot of scientific disciplines is to identify the causes of certain outcome. For example, in new drug development, a personalized medicine strategy can be formed by identifying the genetic markers that is biologically linked to certain drug response or resistance. In electronic health record analysis, the goal is to identify 'actionable' factors for the purpose of reducing cost and improving the quality of care. In epidemiology or sociology studies, it is of interest to identify protection or risk factors, especially those that can be changed through public policy or civil planning to improve public welfare. In these scientific endeavors, the need is to identify the important factors associated with a certain outcome, so that further confirmatory investigation (e.g. randomized international studies) could be conducted to expand knowledge or change future actions for a better outcome. For this purpose, we are interested in procedures that control the Type I error, which is the chance of a false discovery. With a large amount of factors to test, controlling the chance of any false discovery (family-wise error) is too stringent. Therefore the objective is usually relaxed to control the False Discovery Rate (FDR) (Benjamini & Hochberg, 1995).

A recent breakthrough in the statistical theory, i.e. the Model-X framework(Barber & Candès, 2015; Candès et al., 2018), provides a general solution. It can be incorporated with any machine learning models to select true signals associated with the outcome, with rigorous control for FDR. This line of research sheds light on expanding the horizon of supervised learning methods beyond prediction. However, the implementation of Model-X requires the generation of the so called 'knockoffs', which has limited existing methods. The goal of our paper is to fill in the gap by proposing a model-free method for generating knockoffs with relaxed assumptions for the data, leveraging the power of the recent development in deep generative models. Specifically, we propose a framework to generate knockoffs based on latent variable $Z$ which captures the correlation structure of $X$. This provides a great generalization from the existing knockoff generation methods for hidden Markov model (HMM) and mixture Gaussian to (but not limited to) Hidden Markov Random Field. The contribution of this paper is it propose a deep learning framework to generate Model-X knockoffs through latent variable. We provide theoretical justification of the approach and also demonstrate the state of art variational auto encoder (VAE) can achieve promising results in the task of FDR controlled variable selection. This paper is among the first works for the new application of representative learning

in generating model-X knockoffs. Our discussion and preliminary results for goodness-of-fit also shed lights on future improvement of VAE for generating knockoffs.

## 2 BACKGROUND

The model-X framework (Candès et al., 2018) is proposed for the following problem. There are $N$ i.i.d. samples $(X^{(i)}, Y^{(i)})$ from a population, where the predictors are random variables $X = (X_1, \ldots, X_p)$. The outcome $Y$ only depends on a subset of predictors $S \in \{1, 2, \ldots, p\}$, i.e. given $\{X_j, j \in S\}$, $Y$ is independent of $\{X_j, j \notin S\}$. The smallest set $S$ that satisfies this requirement is considered as the true predictors. The goal is to find a good estimator $\hat{S}$ for $S$ with control of the $FDR = E\left(\frac{\#\{j:j \in \hat{S} \setminus S\}}{\#\{j:j \in \hat{S}\} \vee 1}\right)$. Barber & Candès (2015) first proposed the method for linear regression with Gaussian errors, where the design matrix $\boldsymbol{X}$ is assumed to be fixed. Candès et al. (2018) greatly generalized it to the Model-X framework. The traditional statistical models focusing on the conditional distribution $Y|X_1, \ldots, X_p$. In contrast, Model-X makes no assumption for the conditional distribution, but assume $X$ has a known distribution for the purpose of generating knockoffs $\tilde{X}$ that satisfies Definition 2.1.

**Definition 2.1 Model-X knockoffs** *for the family of random variables $X = (X_1, \ldots, X_p)$ are a new family of random variables $\tilde{X} = (\tilde{X}_1, \ldots, \tilde{X}_p)$ constructed with the following two properties: (1) for any subset $S \subset \{1, \ldots, p\}$, $(X, \tilde{X})_{swap(S)} =_d (X, \tilde{X})$; (2) $\tilde{X} \perp\!\!\!\perp Y|X$ if there is a response Y. (2) is guaranteed if $\tilde{X}$ is constructed without looking at Y.*

The following proposition 1 provide an equivalent condition for checking Model-X knockoffs.

**Proposition 1 (Candès 2018)** *The random variables $\tilde{X} = (\tilde{X}_1, \ldots, \tilde{X}_p)$ are Model-X knockoffs for $X = (X_1, \ldots, X_p)$ if and only if the joint distributions are equal after swapping $X_j$ and $\tilde{X}_j$, i.e. $(X_j, X_{-j}, \tilde{X}_j, \tilde{X}_{-j}) =_d (\tilde{X}_j, X_{-j}, X_j, \tilde{X}_{-j})$, for any $j \in \{1, \ldots, p\}$, where $X_{-j}$ denote the other variables. And $Y \perp\!\!\!\perp \tilde{X}|X$.*

Suppose one can generate Model-X knockoffs, the features are selected through the model-specific feature statistics $W_j = w_j([X, \tilde{X}], y)$. The requirement for the feature statistic is that the distribution for the $W_j$'s from the null variables ($X_j \notin S$) need to be symmetric. This is formalized as the flip-sign property,

$$W_j([X, \tilde{X}]_{swap(S)}, y) = (1 - 2\mathbf{1}_{\{j \in S\}})w_j([X, \tilde{X}], y).$$

According to Thm 3.4 in Candès et al. (2018), the following procedure (Knockoff) can estimate the $\hat{S}$ controlling for modified FDR, i.e. $E\left(\frac{\#\{j:j \in \hat{S} \setminus S\}}{\#\{j:j \in \hat{S}\} + q^{-1}}\right) \leq q$. First, compute the threshold $\tau = \min\left\{t > 0 : \frac{\#\{j:W_j \leq -t\}}{\#\{j:W_j \geq t\}} \leq q\right\}$, then the selected set is

$$\text{(Knockoff): } \hat{S} = \{j : W_j \geq \tau\}. \tag{1}$$

A slightly more conservatively procedure (Knockoff+) increase the negative number in numerator by 1, and is able to control for the usual FDR. The threshold for Knockoff+ is $\tau+ = \min\left\{t > 0 : \frac{1 + \#\{j:W_j \leq -t\}}{\#\{j:W_j \geq t\}} \leq q\right\}$, and the selected set is

$$\text{(Knockoff+): } \hat{S} = \{j : W_j \geq \tau+\}. \tag{2}$$

Notice that the Knockoff approach controls for the modified FDR, which might have more power than Knockoff+, but it does not guarantee the control for the usual FDR. In the experiments of this paper, we used the Knockoff+.

An example of the feature statistics is the signed max lambda statistics in $L_1$ penalized regressions Barber & Candès (2015): for the concatenated design matrix $[X, \tilde{X}]$, define $\hat{\beta}(\lambda) = \arg\min_b \left\{Loss(y, [X\tilde{X}]b) + \lambda\|b\|_{L1}\right\}$. Intuitively, the true predictors with strong signals would have entered the LASSO selection with large $\lambda$ and $X_j$ is entering the model with larger $\lambda$ than its

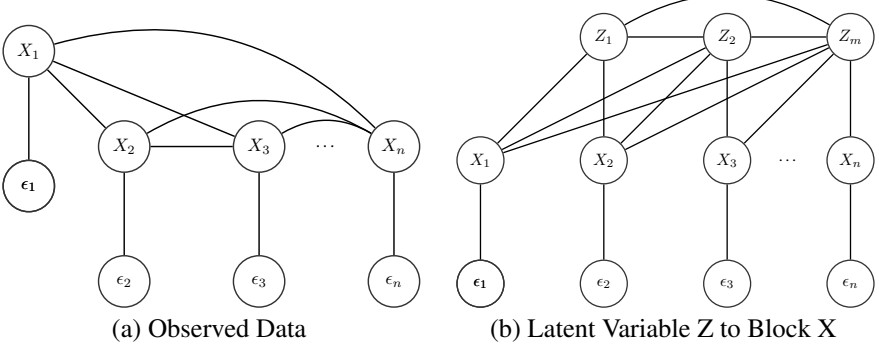

| (a) Observed Data | (b) Latent Variable Z to Block X |

Figure 1: Hidden Markov Field

knockoff $\tilde{X}_j$. The null variables will enter the model with smaller penalty. Thus the signed max lambda statistic is defined as $W_j = (Z_j \vee \tilde{Z}_j)\mathrm{sign}(Z_j - \tilde{Z}_j)$. Where $Z_j = \sup\{\lambda : \hat{\beta}_{2j-1}(\lambda) \neq 0\}$ and $\tilde{Z}_j = \sup\{\lambda : \hat{\beta}_{2j}(\lambda) \neq 0\}$. Various feature statistics has been proposed for common learning methods (Candès et al., 2018; Gimenez et al., 2018), however the current knockoff generation methods are still limited for real data from unknown distributions. Although Candès et al. (2018) provides a general algorithm for generating Model-X knockoffs, i.e. the Sequential Conditional Independent Pairs algorithm, it needs to sample from the conditional distribution of $\tilde{X}_j$ from $f(X_j|X_{-j}, \tilde{X}_{1:j-1})$ which becomes intractable with slightly complex distributional assumption for $X$. Candès et al. (2018) also proposes a second-order approach by matching first two moments of $(X, \tilde{X})_{swap(S)}$ and $(X, \tilde{X})$. The second-order approach generate the true knockoff when $X$ is from Gaussian distribution which have no high-order moments. Sesia et al. (2018) proposes an algorithm to sample knockoffs when $X$ is from Hidden Markov Model. Gimenez et al. (2018) proposes algorithm that applicable for $X$ from simple Bayesian network models (such as mixture Gaussian) with distributional assumptions for certain conditional probabilities which need to be estimated and sampled from.

# 3 A MODEL-FREE KNOCKOFF GENERATOR WITH LATENT ENCODING VARIABLES.

## 3.1 EXISTENCE OF DESIRED LATENT VARIABLE $Z$

Conceptually, the Model-X aims at controlling for the FDR of testing the null hypotheses $X_j \perp\!\!\!\perp Y|X_{-j}$, where $j = 1, \ldots, p$, and $X_{-j}$ denotes the variables except for $X_j$. Thus, a necessary condition for $X_j$ to be chosen by Model-X is that it contains signals not contained in the other variables, i.e. $X_j$ can not be a determinant function of $X_{-j}$. In the knockoff generating process, any component of $X_j$'s that is independent from $X_{-j}$ is easy to be generated. The difficulty and necessity to consider the complex conditional distributions come from the correlated component. Thus we formed a general framework based on the assumption that there exists some latent variable $Z$ so that $X_j$'s are mutually independent conditional on $Z$.

Such $Z$ always exists for any $X$, one can take $Z = X$ or $Z = f(X)$, such that $f$ is an one-on-one mapping between $X$ and $Z$. However, a larger difference in $\tilde{X}_j$ and $X_j$ leads to more powerful knockoffs. In our framework, to generate $\tilde{X}$ different from original $X$, our framework requires there exists $Z$ such that $X|Z$ is mutually independent and $X|Z$ is not degenerated, i.e. $Var(X|Z) > 0$ at least for a subset of $X$ with non-zero measure. In the following, we provide motivational examples followed by a lemma to prove such $Z$ exists for general $X$.

A general class of models satisfies this assumption is the Hidden Markov Random Field as illustrated by the undirected graphical models in Figure 1. The latent variables are from a Markov Random Field and the observed random variables $X_j$'s are generated from $Z's$ with independent errors. However, our method is not limited for Hidden Markov Random Field, $Z$ could exist for the non

Gibbs Random Field ($X$ is not from Markov Random Field). For example, $f(x_1, x_2) = 2\mathbf{1}_{\{x_1 + x_2 \leq 1, x_1 \geq 0 \text{ and } x_2 \geq 0\}}$. Consider the partition of the support $\{x_1 + x_2 \leq 1, x_1 \geq 0 \text{ and } x_2 \geq 0\}$ into countable many squares. The desired $Z$ can be defined as the indicator which squarer $X$ belong to. $X|Z$ is then the uniform distribution on that square, $X_1$ and $X_2$ are independent given $Z$.

The desired latent variable $Z$ always exists for non-degenerate categorical $X$. If $X$ is from binomial (for binary data) or multinomial distribution (for categorical data) with positive probability mass, the desired latent variable $Z$ always exists as a categorical variable with probabilities from the tensor decomposition of multi-way contingency array of probability for $X$ Bhattacharya & Dunson (2012),i.e., $p(x) = \sum_k \pi_k \prod_j p(x_j | z = k)$. Then $X_i|Z$'s are generated from Bernoulli or multinomial with parameters as functions of $Z$ and thus the conditional density $p(X_i|Z)$'s are not degenerate.

For continuous or infinite discrete state $X$, De Finetti's theorem states that if the distribution $(X_1, X_2, \ldots, X_n)$ is exchangeable then there exists a one dimensional non-degenerate latent variable $Z$ that satisfies our requirement. Thus the desired $Z$ exists for the multivariate normal distribution or trans-elliptical distribution (i.e., any distribution that can be transformed into multivariate normal distribution by element-wise transformation). The mixture of trans-elliptical distribution also has the desired latent $Z$ as the combination of the single latent variable which indicate the cluster and the latent $Z$ for each cluster.

More generally, we can prove the desired $Z$ exists with the following Lemma 3.1. In the lemma, we consider $X$ from a non-degenerate distribution with density $f(x)$ with respect to a measure $\mu$ for a sigma field defined on $R^p$. The following lemma showed that one can decompose $f$ into mixture of densities $f_k$, where at least one $f_k$ has non-zero variance.

**Lemma 3.1** *For any non-negative density function $f : R^p \rightarrow R$ whose support contains an interior point, there exist non-negative functions $f_k : R^p \rightarrow R$, $k = 1, 2, \cdots$ such that $f(x) = \sum_k \pi_k f_k(x)$ with $\pi_k > 0$ and $f_k$ satisfies the following: 1)$f_k$'s can be factorized into functions of each coordinates of $x$, i.e., $f_k(x) = \prod_j f_{kj}(x_j)$; 2) at least one component $f_k$ has non-zero variance, i.e. $V(f_k) = \int x^2 f_k(x)\mu(dx) - [\int x f_k(x)\mu(dx)]^2 > 0$.*

**Proof**: Since $f(x)$'s discontinuous points are at most countable, there exists an interior point of the support $x^*$ at which $f(x)$ is continuous. Therefore there exists a cubic area $B(x^*, \epsilon) = \{x : |x_j - x_j^*| \leq \epsilon, j = 1, \cdots, p\}$ such that for any $x \in B(x^*, \epsilon)$, $f(x) \geq f(x^*)/2 > 0$ and $\pi = P(x \in B(x^*, \epsilon)) > 0$. So we have $f(x) = \frac{f(x^*)}{2} \prod_j I(x_j \in [x_j^* - \epsilon, x_j^* + \epsilon]) + g(x)$ where $g(x)$ is a non-negative function. This can be viewed as a mixture of a uniform distribution on the cubic area with probability $\frac{f(x^*)\pi}{2}$ and a distribution with density $\frac{g(x)}{1 - \frac{f(x^*)\pi}{2}}$ with probability $1 - \frac{f(x^*)\pi}{2}$. Notice that the uniform component is mutually independent. The remaining non-negative $g(x)$ can be further decomposed into $\sum_{k=2}^{\infty} \pi_k f_k$, where $f_k$'s are the sequence of step functions to approximate the $g^{-1}(x)$. So now we have $f(x) = \frac{f(x^*)}{2} \prod_j I(x_j \in [x_j^* - \epsilon, x_j^* + \epsilon]) + \sum_{k=2}^{\infty} \pi_k f_k$ and the first term has positive variance, which finish the proof.

**Remark:** The desired $Z$ thus exist as the latent indicator of which component $f_k$, $X$ is from. Where $X|Z$ and $Z$ have the following densities $(X|Z = k) \sim \frac{f_k(x)}{\int f_k(x)\mu(dx)}$ and $P(Z = k) = \pi_k \int f_k(x)\mu(dx)$. At least for the uniform component, $X|Z$ is not degenerated.

To sum up, the desired latent variable $Z$ generally exists for non degenerate continuous and discrete $X$.

## 3.2 METHOD AND ALGORITHM

Our proposed procedure has two steps, the first step is to generate $\tilde{Z}$ from the conditional distribution $Z|X$ and then generate $X$'s knockoff from $X|Z = \tilde{Z}$. Specifically, consider the non-degenerate random vector of covariates $X = (X_1, X_2, \ldots, X_p)$ are from $p_X(x)$. And let $Z$ be the vector of latent random variables $Z$ from distribution $p_Z(z)$. Each component $X_j$ is from $p_{\epsilon_j}(f_i(Z)), j = 1, \ldots, p$, where $\epsilon_j$'s are mutually independent given $Z$. For example, for modeling continuous data, $X_j \sim \mathcal{N}(f_j(Z), \sigma_j^2)$; and for modeling binary data, $X_j \sim \text{Bernoulli}(f_j(Z))$. Given the observed

samples $\{x^{(i)}\}_{i=1}^N$ from $p_X(x)$, to estimate $p_Z(z)$ and $p_{X|Z}$ may not be an identifiable problem. However, we just need to find one $Z$ that approximately satisfies the above condition in practice for our purpose of knockoff generation.

In practice, we assume $Z$ and the conditional distributions are from parametric families and the parameters can be approximated through variational inference. We denote two working models from parametric families: a) an encoder $Q_{Z|X}(z|x;\theta)$ to approximate $p_{Z|X}(z|x)$; b) a decoder $Q_{X|Z}(x|z;f,q_\epsilon)$ to approximate $p_{X|Z}(x|z)$. Where $\theta$ denotes the parameter for the encoder, and $f$ denotes the parameters for decoder. And $q_\epsilon$ denotes the working noise distribution to generate $\tilde{X}$ given $\hat{f}(\tilde{Z})$, and $\epsilon$ is element-wise independent $q_\epsilon = \prod_j q_{\epsilon_j}$. And $\hat{\theta}, \hat{f}$ denote the estimates for these parameters. Then we propose to generate knockoff $\tilde{X}$ as Algorithm 1.

---

**Algorithm 1** Auto-encoding Knockoff Generator.

---

1: Train model parameters to get estimates $\hat{\theta}, \hat{f}$.
2: Generate $\tilde{Z}$ from $Q_{Z|X}(z|X;\hat{\theta})$.
3: Generate $\tilde{X}$ from $Q_{X|Z}(x|\tilde{Z};\hat{f},q_\epsilon)$.

---

Notice that the Step 2 in Algorithm 2 is different from the usual VAE implementation. In the latter, $\tilde{Z}$ is usually generated from a prior distribution $p_Z(z)$, however in our algorithm $\tilde{Z}$ is generated from the conditional distribution $Q_{Z|X}$. The following Theorem 3.2 justifies why our procedure is so.

By fitting a working model from parametric families with $X|Z$ mutually independent, the good reconstruction of $X$ from this model is a surrogate objective for $\tilde{X}$ is the knockoff of $X$. Thus the state of art Variational Auto Encoders (VAEs) (Kingma & Welling, 2014; Rezende et al., 2014; Maddison et al., 2017; Jang et al., 2017) provide a way to estimate the working models in Step 1. It also worth noting that Gao et al. (2018) point out that under certain conditions, the ELBO for VAE is equivalent to a lower bound for the Total Correlation Explanation, which is optimized when $Z$ fully decorrelate (disentangle) the $X$ (i.e. conditional of Z, X is mutually independent).

We explain the connection between our notation and the VAE methods in the following two examples. In the common variational auto encoder, $Z \sim N(0, I_q)$, the working model $Q_{Z_i|X}$ is assumed to be from $\mathcal{N}(\mu_i(X), \sigma_i^2(X))$, and $Q_{X_i|Z}$ is $\mathcal{N}(f_i(Z), \sigma^2)$, where $\mu_i(\bullet)$ and $\sigma_i^2(\bullet)$ and $f(\bullet)$ are functions to be approximated by neural networks. The $q_\epsilon$ in this case is i.i.d. normal distribution with infinitesimal variance. So in practice people generate $\tilde{X}$ as $\hat{f}(\tilde{Z})$. Another example is to approximate discrete $Z$'s by the concrete distribution Jang et al. (2017); Maddison et al. (2017). In this case $\tilde{X}$ is generated from $Bernoulli(\hat{f}(\tilde{Z}))$ and $\epsilon = \tilde{X} - \hat{f}(\tilde{Z})$ are element-wise independent.

**Remark:** The objective function of our learning framework targets directly on the reconstruction of $X$, the data's compatibility with the working models for $Z|X$ and $X|Z$. Alternatively, if one construct the knockoffs directly from deep neural network to approximate $\tilde{X} = f(X, E)$ where $E$ represents the generated random noise. The objective is then to minimize the the discrepancy between two distributions $(X, \tilde{X})$ and $(X, \tilde{X})_{swap(S)}$. Since $S$ is any subset of $\{1,\ldots, p\}$, there are $2^p - 1$ ways of swapping. Thus the objective should be a composite function that incorporate all the swapping. This approach may have advantage over our approach when the 'working' variational auto encoding model is not compatible with the true data distribution. Our proposed framework may have advantage in less computational burden and easiness to implement. The pros and cons of our method and various other deep learning based knockoff generator await to be evaluated and compared case by case in various real data sets.

### 3.3 THEORETICAL RESULTS

Theorem 3.2 shows that if X is mutually independent conditional on $Z$, if given the true conditional distributions $Z|X$ and $X|Z$, one can first generate $\tilde{Z}$ from $Z|X$ and then generate $\tilde{X}$ from $X|Z = \tilde{Z}$. The $\tilde{X}$ is the model-X knockoff of $X$. Theorem 3.3 provides the FDR bound for estimated conditional distribution $Z|X$ and $X|Z$.

**Theorem 3.2** *For any vector of random variables $Z$, such that conditional on $Z$, $X_i$'s are mutually independent. If $\tilde{Z}$ is from $p_{Z|X}(\bullet|X)$, then $\tilde{X}$ from $p_{X|Z}(\bullet|\tilde{Z})$ is model-X knockoffs.*

**Proof:** For any $j \in 1, \ldots, p$ and sets $A_j, B_j, A_{-j}, B_{-j}$,

$$P(X_j \in A_j, \tilde{X}_j \in B_j, X_{-j} \in A_{-j}, \tilde{X}_{-j} \in B_{-j}) = \int p_{X|Z}(A_j, A_{-j}|Z)p_{X|Z}(B_j, B_{-j}|Z)dF(Z)$$

$$= \int p_{X_j|Z}(A_j|Z)p_{X_{-j}|Z}(A_{-j}|Z)p_{X_j|Z}(B_j|Z)p_{X_{-j}|Z}(B_{-j}|Z)dF(Z)$$

$$= \int p_{X_j|Z}(B_j|Z)p_{X_{-j}|Z}(A_{-j}|Z)p_{X_j|Z}(A_j|Z)p_{X_{-j}|Z}(B_{-j}|Z)dF(Z)$$

$$= \int p_{X|Z}(B_j, A_{-j}|Z)p_{X|Z}(A_j, B_{-j}|Z)dF(Z) = P(X_j \in B_j, \tilde{X}_j \in A_j, X_{-j} \in A_{-j}, \tilde{X}_{-j} \in B_{-j})$$

We have shown $X_j$ and $\tilde{X}_j$ are exchangeable in the joint distribution, then $\tilde{X}$ is the Model-X knockoff by Proposition 1.

Now we consider the FDR bound for the knockoffs generated from the estimated encoder and decoder. We assume the working models and observed distribution of $X$ are approximately compatible as in the conditions of the following theorem. We then apply the results per Barber et al. (2018) to show FDR control, where they showed the FDR derived from approximate knockoffs is bounded by a function of the observed KL divergence between $(X_j, \tilde{X}_j, X_{-j}, \tilde{X}_{-j})$ and $(\tilde{X}_j, X_j, X_{-j}, \tilde{X}_{-j})$ as below:

$$\widehat{KL}_j = \log\left(\frac{P(X_j, \tilde{X}_j, X_{-j}, \tilde{X}_{-j})}{P(\tilde{X}_j, X_j, X_{-j}, \tilde{X}_{-j})}\right) = \sum_{i=1}^{n} \log\left(\frac{P(X_{ij}, \tilde{X}_{ij}, X_{i,-j}, \tilde{X}_{i,-j})}{P(\tilde{X}_{ij}, X_{ij}, X_{i,-j}, \tilde{X}_{i,-j})}\right)$$

**Theorem 3.3** *Assume the observed marginal distribution $p_X(x)$ and the working models ($Q_{Z|X}$, $Q_{X|Z}$, $q_\epsilon$ estimated in Algorithm 1) are approximately compatible as following: there exists a random vector $Z$ from certain distribution $p_Z(z)$ and $a_n \to 0$, such that*

$$\sup_x |\log\left(\frac{q_\epsilon(x)}{p_{\hat{\epsilon}}(x)}\right)| \leq a_n \text{ and } \sup_{z,x} |\log\left(\frac{p_{Z|\hat{X}}(z|x)}{Q_{Z|X}(z|x)}\right)| \leq a_n.$$

*Where the density of $\hat{\epsilon} = X - \hat{f}(Z)$ is denoted as $p_{\hat{\epsilon}}$, considering $\hat{f}$ as a fixed function. And denote $\hat{X} = \hat{f}(Z) + \epsilon$ as a random variable generated from the decoder $Q_{X|Z}(x|z = Z; \hat{f}, q_\epsilon)$.*

*Then the FDR can be controlled at $q \exp\{8na_n^2 + 8\sqrt{n \log(p)}a_n\}$.*

Proof of Theorem 3.3 is in Appendix A.

**Remark:** Theorem 3.3 provides FDR bound given the variational models is able to approximate the $Z|X$ and $X|Z$ sufficiently close. However, there is no guarantees for deep generative model to converge with the conditions as in Theorem 3.2. And generally, deep learning knockoff generators gain the flexibility of no distributional assumptions at the cost of no current theoretical guarantee directly from convergence of algorithm to control of FDR. We will discuss this further in section 7. And we provide two ways for evaluating knockoff generated from our proposed framework in the following section 4.

## 4 GOODNESS OF FIT

One evaluation approach proposed by Candès et al. (2018) is to estimate the empirical FDR for simulated $Y$ from the $X$ of the real data, and compare the power and FDR control performance of knockoffs from various generator. Another method is to compose some goodness of fit metrics based on the distributional discrepancy between $(X, \tilde{X})$ and $(X, \tilde{X})_{swap}$. We will discuss these metrics further in section 7. In addition to the metrics shared with other knockoff generator, our framework has a unique measurement of goodness of fit based on the mutual independence condition

for $X$ given $Z$. Specifically, if the $q_\epsilon$ is assumed to be additive error, i.e. $\tilde{X}_j = \hat{f}_j(\tilde{Z}) + \epsilon_j$. The mutual independence of the residuals $X - \hat{f}(\tilde{Z})$, which is the difference of the real data $X$ and its fitted means, indicate good model fitting. In the following numerical studies, we focus on the first evaluation strategy which is based on FDR control for simulated $Y$. We include preliminary graphical check for the independence criterion in Appendix C and further discussion in Section 7.

## 5 SIMULATION

In the simulation, we compare three knockoff generation schemes: a) our proposed method implemented through VAE b) the fixed knockoff generation method in Barber & Candès (2015) c) the second order matching in Candès et al. (2018). We demonstrate the performance of these methods in two simulation scenarios. In both scenarios, the sample size is 200, and number of potential predictors is 100, where the first $m$ variables are the true signals. The coefficients $\beta$ for the true predictors are alternating $\rho$ and $-\rho$, where $\rho$ is the magnitude of the signal. In Figures 23, the power and FDR are drawn as curves with respect to $\rho$.

We generated two types of outcomes: the continuous outcomes are generated with errors from standard normal distribution; the binary outcomes are generated from Bernoulli with probabilities $1/(1 + \exp(-X\beta))$. We use the signed max lambda statistics in Barber & Candès (2015) (see definition on page 2). We implemented the $L_1$ penalized linear and logistic regression with R package 'glmnet' correspondingly. We used the Knockoff+ as in (2) to control for FDR. Results based on 100 replications are presented for two different numbers of true signals $m = 10, 20$.

**Setting 1.** In this setting, we simulate a set of lightly correlated non-normal distributed continuous predictors: a) Generate independent uniform $(0, 1)$ distributed 200 by 200 random matrix. b) Let C be the Cholesky decomposition of the correlation matrix with $0.1$ on the off diagonal entries and compute $Z = UC$. c) Even column $X_{2i}$ is generated as $Z_{4i} + 0.5 * Z_{4i+1}^3$, and odd column $X_{2i+1}$ is generated as $Z_{4i+2} - 0.5Z_{4i+2}^2 + 0.5 \exp(Z_{4i+3})$, where $i = 0, \ldots, 49$. d) Rescale $X$'s to be within range $[0, 1]$ by subtracting the column $\min$ and divided by column $\max - \min$. Empirically, the pairwise correlation $X$ ranges from $-0.3$ to $0.6$ and the absolute value of the correlation has mean $0.07$. The signal to noise ratios of the Gaussian case approximately has the range of $0.25 - 0.3\rho^2$ for $m = 10$ and $0.7 - 0.85\rho^2$ for setting with $m = 20$.

**Setting 2.** The second setting generates correlated categorical predictors :a) Generate independent standard Gaussian distributed 200 by 100 random matrix. b) Let C be the Cholesky decomposition of the correlation matrix with $0.1$ on the off diagonal entries. Let $Z = UC$. c) $X$ is the categorized $Z$ matrix, where $X_{ij} = \mathbf{1}(Z_{ij} > 0)$. Approximately, the signal to noise ratio for the Gaussian case is $2\rho^2$ for $m = 10$ and $4\rho^2$ for $m = 20$. We present results for FDR controlled at $0.1$.

**Results** According to Figures 2 3, our proposed method implemented through VAE has FDR controlled below the predetermined threshold, and it demonstrates higher power than its two competitors. Cases with more true signals $m = 20$ demonstrate better power for Gaussian outcomes, since we control for the proportion of false discoveries, the criterion can tolerate more false discoveries with larger number of true discoveries it detect. The second-order matching method proposed for the gaussian predictors has very low power due to the violation of its assumption that our generated $X$ are not from Gaussian distribution. The assumptions for Fixed Knockoff generator is satisfied for the linear regression settings(with gaussian errors) but violated for the logistic regression setting. In the simulation, the FDR is still controlled for cases these assumptions are violated. However the powers are low.

**Implementation details.** For both settings, we implemented our method with VAE Kingma & Welling (2014). We defer the training details to Appendix B.

## 6 REAL DATA EXPERIMENT

In the real data experiment, we demonstrate the performance of our proposed method on generating knockoffs for sparse genetic mutation data. The dataset and scientific problem is from a study aiming at detecting mutations associated with drug resistance in patients with Human Immunodeficiency Virus Type 1 (HIV-1) Rhee et al. (2006). Due to the space limit, here we present results

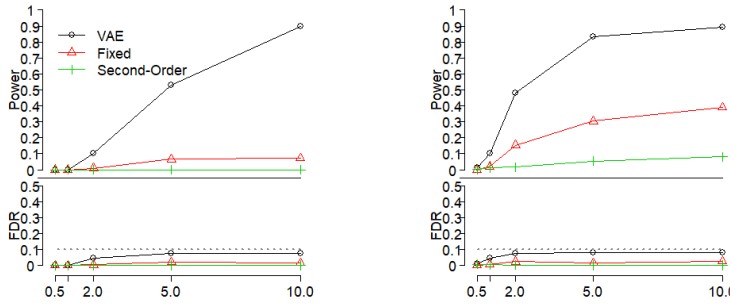

(a) Linear Regression $m = 10$, FDR= 0.1    (b) Linear Regression $m = 20$, FDR= 0.1

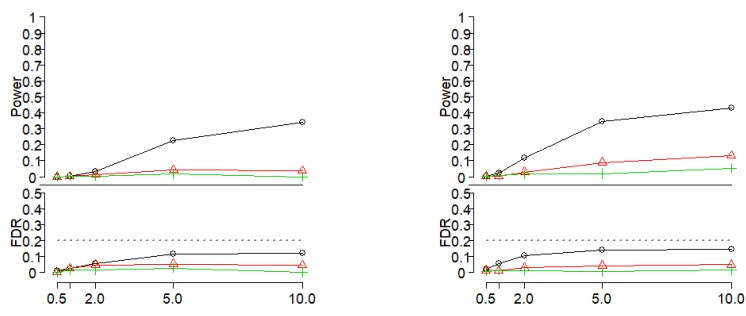

(c) Logistic Regression $m = 10$, FDR= 0.2   (d) Logistic Regression $m = 20$, FDR= 0.2

Figure 2: Power and FDR with respect to signal magnitude $\rho$ for Setting 1.

for the 7 protease inhibitors. There is no ground truth for which subset of mutations caused the drug resistance. Nevertheless, there is a set of 73 treatment-selected mutations (TSMs) identified from a separate study Rhee et al. (2005), which are selected as the mutations marginally correlated with the patient treatment history with protease inhibitors. Thus to evaluate the performance of the knockoffs, we first simulates the outcome for which we know the true signal, and evaluate the FDR control and power as shown in Figure 4, and then we used the real data outcome and compared our selected mutations with the TSM. Notice that the TSM is not drug specific and can not be considered as ground truth, by showing number of selected mutations in TSM set we are not showing the FDR control but demonstrate reproducibility across studies.

The normal distributed $Z$ does not fit the discrete and sparse nature of the genetic mutation data. Thus we adopted the Categorical Variation Auto-Encoder (CAT-VAE) Jang et al. (2017); Maddison et al. (2017), where in our implementation, the latent variables $Z$ are 20 Gumbel-Softmax distributed variable with temperature 1, each with 10 categories; both the encoder and decoder has one hidden layer of dimension 200.

Figure 4 demonstrates the results for simulated outcomes using the real data $X$. Overall speaking, our proposed method still achieves the FDR control and demonstrates the highest power among the three. The correlation is smaller than the previous simulation settings (since the X is a sparse matrix), so the other two methods has better power. Controlled for a FDR level of 0.2, all three methods achieve a high power of 80% very large signal at $\rho = 10$ in the Gaussian case. For the binary case, our proposed method shows greater advantage in achieving more than 2-times the power of the other two methods.

Table 1 presents the number of selected and matched selection of mutations with the 73 TSMs. Here we present results controlled by both Knockoff+ (as in (2)) and Knockoff (as in (1)) controlled at FDR level 0.2. Similar to the simulated settings, our proposed method select more mutations

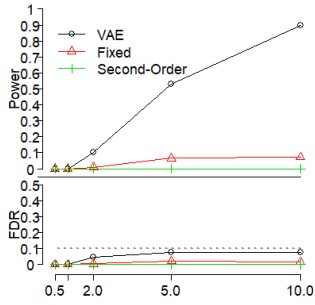 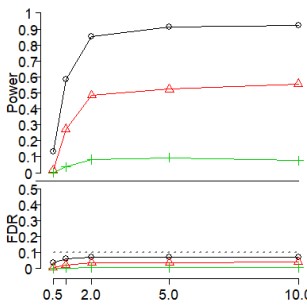

(a) Linear Regression $m = 10$, FDR= 0.1    (b) Linear Regression $m = 20$, FDR= 0.1

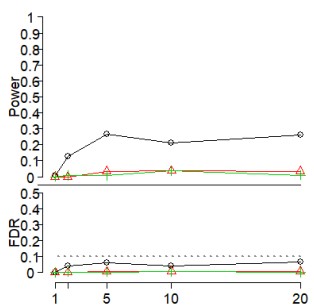 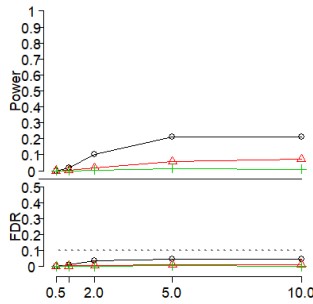

(c) Logistic Regression $m = 10$, FDR= 0.1 (d) Logistic Regression $m = 20$, FDR= 0.1

Figure 3: Power and FDR with respect to signal magnitude $\rho$ for Setting 2.

Table 1: Number of TRM mutations/ total number of selected mutations controlled for FDR at 0.2

|  |  | APV | ATV | IDV | LPV | NFV | RTV | SQV |
|---|---|---|---|---|---|---|---|---|
|  | Sample Size | 767 | 328 | 825 | 515 | 842 | 793 | 824 |
| Knockoff+ | CAT-VAE | 28/28 | 8/31 | 34/49 | 27/37 | 42/54 | 38/48 | 39/52 |
|  | Fixed | 25/25 | 2/10 | 0 | 0 | 0 | 35/44 | 23/25 |
|  | Second-Order | 0 | 0 | 0 | 0 | 5/5 | 5/5 | 0 |
| Knockoff | CAT-VAE | 32/40 | 9/38 | 34/49 | 28/38 | 42/54 | 38/48 | 38/50 |
|  | Fixed | 30/35 | 0 | 14/14 | 0 | 26/26 | 34/43 | 25/27 |
|  | Second-Order | 24/24 | 4/11 | 4/4 | 17/24 | 29/29 | 35/42 | 17/17 |

than the other methods, the average proportion of selected mutation that is in the TSM set for our proposed method is 75.3%, which demonstrate reproducibility across studies. With less conservative Knockoff procedure, Fixed and Second-Order methods are able to select more mutations, where our proposed method seems be less sensitive to choice of Knockoff or Knockoff+.

**Implementation details** For these analysis, the CAT-VAE knockoffs are generated for 186 mutations with ($\geq 4$) occurrence in the data set and the number of patients is 846. Since each drug resistance outcome is observed in a subset of samples, when conduct analysis for each drug, we only consider mutations with $\geq 2$ occurrence. Following analysis in Rhee et al. (2006), the outcome is a continuous variable, which is the log transformed drug susceptibility. For results in Figure 4, the linear signal is simulated as following: in each replication, randomly chose 50 predictors from 186 mutations , and compute $\mu$ as sum of $\rho x_j$'s with alternating signs. The Gaussian outcome is $\mu$ plus a standard normal noise and the binomial outcome is generated from Bernoulli($\frac{1}{1+\exp(-\mu_i)}$).

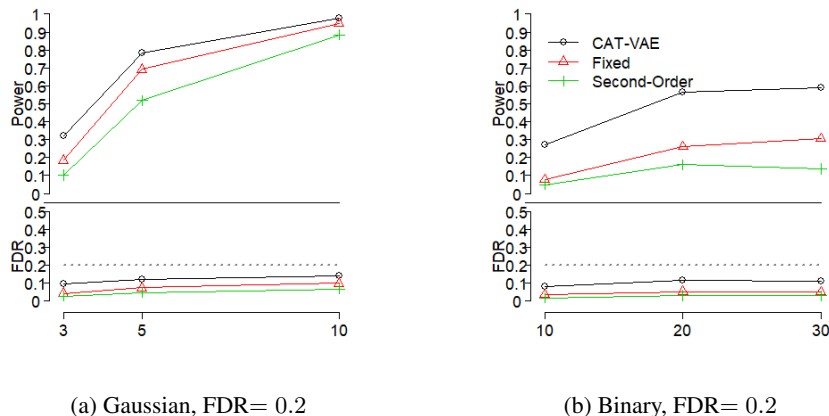

(a) Gaussian, FDR= 0.2                    (b) Binary, FDR= 0.2

Figure 4: Capture of simulated signals with real data generated knockoffs

## 7 CONCLUSION AND DISCUSSION

In this paper, we propose a knockoff generating framework based on a latent variable $Z$ to decorrelate $X$. We noticed there are two independent works on deep learning generators for knockoffs while writing and revising the paper. (For double blind review consideration, we exclude the citations at this time.) Compared with the other deep learning frameworks for generating knockoffs, our framework is the only framework directly targeting on the reconstruction of $X$ from $Z$. We provide theoretical results for general existence of such $Z$. Our framework offers a computational light solution, and if the data and the parametric families for the working models are compatible, our framework may offer a computationally efficient solution to be easily implemented by domain scientists.

Since deep learning approaches mainly target on real data from complex distributions, which does not have tractable conditional distributions. There is a lack of theoretical FDR guarantee for all the current deep knockoff generators. One common obstacle is the non-convexity of deep learning objectives. To solve this problem, an arXiv paper (name omitted for double blind review) proposed several goodness of fit statistics to compare deep generative models. It calls for future research to compare these deep knockoff generators in terms of their power and FDR control performance in simulated $Y$'s and also compare them via various goodness-of-fit metrics.

In addition to these metrics, the structure of our framework provides an unique diagnostic metric, which is the mutual independence of $X|Z$ as described in section 4. We include the some result for graphical check in Appendix C, which demonstrates the estimated $Z$ does attenuate $X$'s correlation. In both simulation settings and the real HIV dataset, the attenuation for the binary $X$ is closer to independence compared with the continuous simulated $X$. It remains for further investigation that if we can adjust the objective function of VAE to improve the goodness-of-fit in terms of independence. For example, adding a penalization term with the Maximum Mean Discrepancy metric between distribution of $X - f(\tilde{Z})$ and the element wise independent distribution from the product of marginal distributions.

To sum up, this paper proposes a framework of deep knockoff generator based on latent variables. This framework offers a computationally light solution to generate knockoffs. The promising numerical results and the easiness of implementation show make it a promising tool among the choices of deep generating models for FDR controlled variables selection.

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

## A. PROOF OF THEOREM 3.2

**Proof:** Consider $\tilde{\hat{X}}$ which is another independent sample from $Q_{X|Z}(x|z = Z; q_\epsilon)$ when sampling $\hat{X}$ using the same $Z$. Since $q_\epsilon$ is element-wise independent, conditions in Theorem 2.1 holds, thus $\tilde{\hat{X}}$ is a model-X knockoff of $\hat{X}$ and

$$\log\left(\frac{P(\hat{X}_j \in A_j, \tilde{\hat{X}}_j \in B_j, \hat{X}_{-j} \in A_{-j}, \tilde{\hat{X}}_{-j} \in B_{-j})}{P(\hat{X}_j \in B_j, \tilde{\hat{X}}_j \in A_j, \hat{X}_{-j} \in A_{-j}, \tilde{\hat{X}}_{-j} \in B_{-j})}\right) = 0.$$

Notice that by our assumptions

$$\sup_x |\log\left(\frac{p_{\hat{X}}(x)}{p_X(x)}\right)| \leq \sup_{x,z} |\log\left(\frac{p_{\hat{X}|Z}(x|z)}{p_{X|Z}(x|z)}\right)| = \sup_{x,z} |\log\left(\frac{q_\epsilon(x - \hat{f}(z))}{p_{\hat{\epsilon}}(x - \hat{f}(z))}\right)| \leq a_n$$

$$\sup_{x,\tilde{x}} |\log\left(\frac{p_{\tilde{X}|\tilde{X}}(\tilde{x}|x)}{p_{\tilde{X}|X}(\tilde{x}|x)}\right)| = \sup_{x,z,\tilde{x}} |\log\left(\frac{\int q_\epsilon(\tilde{x}-\hat{f}(z))p_{Z|\hat{X}}(z,x)dz}{\int q_\epsilon(\tilde{x}-\hat{f}(z))Q_{Z|X}(z,x)dz}\right)| \le \sup_{z,x} |\log\left(\frac{p_{Z|\hat{X}}(z|x)}{Q_{Z|X}(z|x)}\right)| \le a_n$$

So $\sup_{x,\tilde{x}} |\log\left(\frac{p_{(\hat{X},\tilde{\hat{X}})}(x,\tilde{x})}{p_{(X,\tilde{X})}(x,\tilde{x})}\right)| \le 2a_n$ and we can bound the log likelihood ratio by

$$\log\left(\frac{P(X_j \in A_j, \tilde{X}_j \in B_j, X_{-j} \in A_{-j}, \tilde{X}_{-j} \in B_{-j})}{P(X_j \in B_j, \tilde{X}_j \in A_j, X_{-j} \in A_{-j}, \tilde{X}_{-j} \in B_{-j})}\right)$$

$$= \log\left(\frac{P(\hat{X}_j \in A_j, \tilde{\hat{X}}_j \in B_j, \hat{X}_{-j} \in A_{-j}, \tilde{\hat{X}}_{-j} \in B_{-j})}{P(\hat{X}_j \in B_j, \tilde{\hat{X}}_j \in A_j, \hat{X}_{-j} \in A_{-j}, \tilde{\hat{X}}_{-j} \in B_{-j})}\right) + \log\left(\frac{P(X_j \in A_j, \tilde{X}_j \in B_j, X_{-j} \in A_{-j}, \tilde{X}_{-j} \in B_{-j})}{P(\hat{X}_j \in A_j, \tilde{\hat{X}}_j \in B_j, \hat{X}_{-j} \in A_{-j}, \tilde{\hat{X}}_{-j} \in B_{-j})}\right)$$

$$+ \log\left(\frac{P(\hat{X}_j \in B_j, \tilde{\hat{X}}_j \in A_j, \hat{X}_{-j} \in A_{-j}, \tilde{\hat{X}}_{-j} \in B_{-j}))}{P(X_j \in B_j, \tilde{X}_j \in A_j, X_{-j} \in A_{-j}, \tilde{X}_{-j} \in B_{-j})}\right) \le 0 + 2a_n + 2a_n = 4a_n$$

So we can bound

$$\max_{j=1,\cdots,p} \widehat{KL}_j \le 8na_n^2 + 8\sqrt{n\log(p)}a_n.$$

And FDR has the bound in the statement by applying Lemma 2 in Barber et al. (2018). Specifically, if $a_n = o((n\log(p))^{-1/2})$, then FDR will be asymptotically bounded by $q + o(1)$.

## B. IMPLEMENTATION DETAILS FOR NEURAL NETWORKS IN SIMULATION

The latent variables are multivariate normal with 300 dimensions. The models were trained with batch size 25 and for 20 epochs with the default 'adam' optimizer in 'keras'Chollet et al. (2015). The architecture of VAE for setting 1 is as following, the encoder networks for $\mu_z$ and $\log(\sigma)$ have two hidden layers with 500 and 400 neurons with 'tanh' activation and $L_2$ regularization with tuning parameter 0.2. The activation for the output layer is 'linear'. The decoder network has two hidden layers with the same specifications, followed by a batch-normalization layer and output with linear activation. The architecture of VAE for setting 2 is as following, the encoder networks has two hidden layers with 500 and 400 neurons with 'relu' activation and $L_2$ regularization with tuning parameter 0.3. The activation for the output layer is 'linear'. The decoder is an one layer network with the 'sigmoid' activation function and then threshold by 0.5.

## C. ILLUSTRATION FOR DISENTANGLEMENT OF $Z$.

Here we present the Q-Q plots for the p-values for pairwise correlation test for $X$ and $X|Z$. Where the correlation of $X|Z$ is estimated as the pairwise correlation of $X - \hat{f}(\tilde{Z})$. When $X|Z$ are mutually independent, the p-values should be from the uniform distribution and the Q-Q plot will coincide with the diagonal line $y = x$. For the simulated settings, the figure is the result from one replication.

We see for the HIV real data example in Figure A1 (e) (f), the correlation is greatly attenuated by conditioning on $Z$. For the simulation studies, the Q-Q plot in setting 2 (Figure A1 (c) and (d)) almost coincide with the diagonal line after conditioning on $Z$, which indicates good fit of the model. Interestingly, in the above two settings, $X$'s are binary. In simulation setting 1 (Figure A1 (a) and (b)), when $X$'s are continuous, the disentanglement effect is still visible but not as strong as setting 2 and the HIV example.

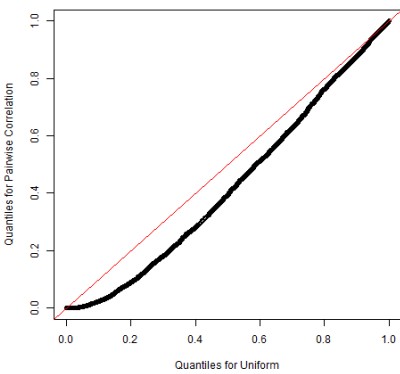

(a) Q-Q plot for p-values of correlation of $X$ in simulation setting 1

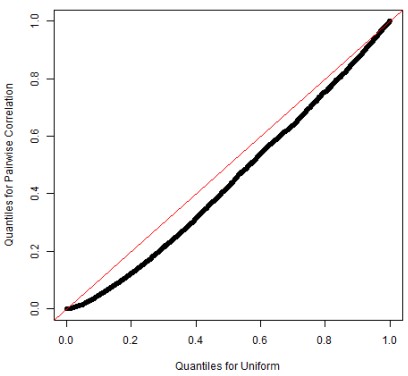

(b) Q-Q plot for p-values of correlation of $X - \hat{f}(\tilde{Z})$ in simulation setting 1

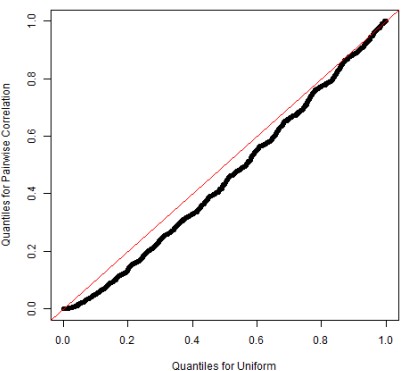

(c) Q-Q plot for p-values of correlation of $X$ in simulation setting 2

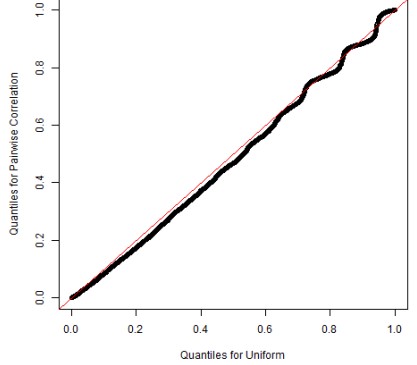

(d) Q-Q plot for p-values of correlation of $X - \hat{f}(\tilde{Z})$ in simulation setting 2

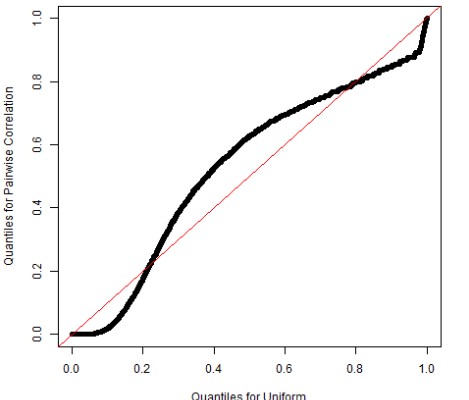

(e) Q-Q plot for p-values of correlation of $X$ in the HIV example

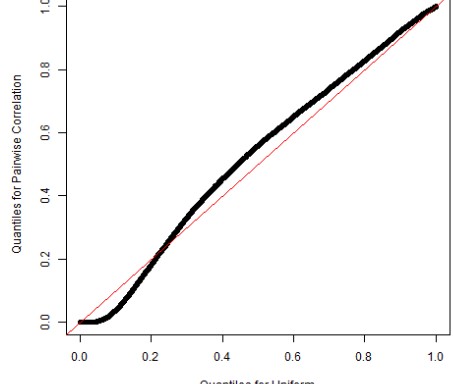

(f) Q-Q plot for p-values of correlation of $X - \hat{f}(\tilde{Z})$ in the HIV example

Figure A1. Illustration of Attenuation Effect of $Z$ for $X$'s Correlation.

