# OpenReview forum: "Auto-Encoding Knockoff Generator for FDR  Controlled Variable Selection"
_ICLR.cc/2019/Conference_

### Official Review · AnonReviewer1 · 2018-10-24
**Good results, but a slight gap between theory and practice.**

**Rating:** 6
**Confidence:** 3

**Review:**

In the paper , the authors proposed the use of autoencoder for Model-X knockoffs. The authors proved that, if there exists latent factors, and if the encoders and the decoders can approximate conditional distributions well, the autoencoder can be used for approximating Model-X knockoff random variables: one can find relevant features while controlling FDR (Theorem 2.2).

I think the theoretical part is good, and the experimental results seem to be promising.

My concern is the gap between theory and practice. In the manuscript, the authors used VAE for approximating conditional distributions. The question is how we can confirm that the trained encoder and decoder satisfy the assumptions in Theorem 2.2. If the trained models violate the assumptions, the control of FDR is no longer guaranteed, which may lead to false discoveries. As long as this gap remains unfilled, we cannot use the procedure reliably: we always need to doubt whether the encoders and decoders are trained appropriately or not. I think this gap is unfavorable for scientific discovery where only rigid procedures are accepted.
How we can use the proposed procedure reliably, e.g. for scientific discovery? Is there any way to confirm that the encoders and decoders are appropriate? Or, is there any way to bypass the gap so that we can guarantee the FDR control even for inappropriate models?

---

> ### Author Response · Authors · 2018-11-19
> **Revision uploaded to discuss more about the gap between theory and practice**
>
> Thanks for the positive feedback.
>
> The theorem 3.3 (previous 2.2) although provide a bound for FDR, but it has limited implication for practice because the lack of general theoretical convergence rate for VAE.
> This is a common problem in all the emerging deep learning knockoff generators.
>
> To fill in the gap between current theory and practice, we provide more explanation and discussion regarding how to validate and evaluate the knockoffs generated from our framework in section 4, section 7, and appendix C. Notice that in the previous version, we include some of the discussion in the discussion section.  In this revision we expand this in to a more comprehensive description and  discussion, we also include some numerical results in Appendix C.
>
> There are two approaches to validate the knockoffs.  One evaluation approach proposed by Candes 2018,  is to estimate the empirical FDR for simulated $Y$ and $X$ of the real data, and compare the power and FDR control performance of knockoffs from various generator. Then it will provide confirmation for domain scientist that if the true signal is similar to the simulated settings, the FDR is controlled.
>  The other approach proposed by a most recent preprint (omitted the name for double blinded review purpose), is to compose some goodness of fit metrics based on the distributional discrepancy between $(X,\tilde{X})$ and $(X,\tilde{X})_{swap}$.   Both approaches offer tools to compare and choose the best knockoff generator for each specific real data applications. To further fill in the gap between theory and practice, new Robust knockoff theories need to be further investigate to link these empirical goodness of fit metrics with bound to FDR.
>
> And in addition to the metrics shared with the other knockoff generators, our framework has an unique measurement for goodness of fit, which is that $X|Z$ are independent. Specifically, if the $q_\epsilon$ is assumed to be additive error, i.e. $\tilde{X_j} = \hat{f_j}(\tilde{Z})+\epsilon_j$. The mutual independence of the residuals $X-\hat{f}(\tilde{Z})$, which is the difference of  the real data $X$ and its fitted means,  indicate good model fitting.
>
> Please refer to the revised section 3, 4, 7 and appendix C. for  expanded explanation of this point.
>
> Thanks,

---

### Official Review · AnonReviewer2 · 2018-10-31
**A reasonable but unfortunately flawed approach to FDR control in feature selection combining neural networks and the knockoff**

**Rating:** 4
**Confidence:** 4

**Review:**

This manuscript tackles an important problem, namely, generating the knockoff procedure for FDR-controlled feature selection so that it can work on any data, rather than only for data generated from a Gaussian (as in the original work) or several specific other cases (mixtures of Gaussians or HMMs).  The basic idea is to exploit ideas from variational autoencoders to create a generic knockoff generation mechanism. Specifically, the main idea of the paper is to map the input covariates X into latent variable Z using a variational auto-encoder, generate the knockoffs \tilde{Z} in the latent space, and then map \tilde{Z} back to the input space to get the knockoffs \tilde{X}. The authors claim that their contribution in threefold:

(1) Given the assumption is valid that X is mutually independent conditional on Z, the generated knockoffs \tilde{X} is proved to be valid in terms of satisfying the necessary swap condition.

(2) Given (1) holds, and given that the discrepancy (measured by KL-divergence) between the true conditional probability Q(Z|X) and its estimate using variational auto-encoder is bounded by o((nlogp)^{-1/2}), the FDR is also bounded.

(3) The proposed knockoffs generating procedure can achieve controlled FDR and better power.

In agreement with the above intuition, I have major concerns about the correctness of the paper.

The cornerstone of the proof in contribution (1) relies on the assumption that X is mutually independent conditional on Z. However, this assumption is invalid if there are dependencies between x_i and x_j. Therefore, the proposed knockoffs \tilde{X} cannot be proved valid by Theorem 2.1.

The erroneous proof in contribution (1) leads to the failure of contribution (2) stated in Theorem 2.2. The FDR is no longer controlled. Intuitively, according to algorithm 1, \tilde{Z} and \tilde{X} are expected to have the same distribution as Z and X, respectively; therefore, the FDR cannot be controlled.

The experimental results are suspicious. In general, it seems fishy that the proposed VAE approach outperforms Model X in the fully Gaussian case.  In this setting, Model X should have an advantage, since it is specifically designed for Gaussian generated data.  Related to this, the text is confusing: "Since the data were not Gaussian, the second-order matching method has the lowest power. The assumptions of the Fixed knockoff generations holds for the Gaussian cases, …" In the figure, the second-order matching method has low power even in the Gaussian case.

Minor comments:

p. 2: The exposition should explain the motivation for Knockoff+.

The manuscript contains numerous grammatical errors, a few examples of which are below:

p. 1: "biological linked" -> "biologically linked"

p. 1: "associated certain" -> "associated with a certain"

p. 1: "showed light" -> "shed light"

p. 1: "which has very limited" -> "which has limited"

p. 1: "leveraging on the power of of" -> "leveraging the power of"

---

> ### Author Response · Authors · 2018-11-06
> **Clarification**
>
> Thanks for the review and many constructive suggestions. I will write the formal point to point response latter. However, I found there is a major concern for the theoretical results about the proof of contribution (1) that I want to discuss ahead and  to make sure we understand the concern.
>
> "The cornerstone of the proof in contribution (1) relies on the assumption that X is mutually independent conditional on Z. However, this assumption is invalid if there are dependencies between x_i and x_j. Therefore, the proposed knockoffs \tilde{X} cannot be proved valid by Theorem 2.1."
>
> Even if two variables X_i and X_j are not independent. Conditional on Z, X_i and X_j can be independent.
> For example, Z is a standard normal variable,  e1 and e2 are two independent error variables (e1 and e2 are independent, and they are independent of Z).  Generate  X_1=Z+ e1, X2=Z+e2.  Then X_1 and X_2 are independent conditional on Z.
>
> Our proposed method relies on 'X_i is independent of X_j conditional on Z'.    Indeed we are using the VAE for approximate the latent Z that de-correlates X's.  There are some works in pre-prints that actually point out in the informatics theory, VAE is targeting to find the Z that de-correlates X. https://arxiv.org/pdf/1802.05822.pdf
>
> Does this clarification solve the concern about 1)?
>
> Admittedly, we should move the proofs to the appendix and so to have  a longer introduction and results discussion. Sorry for the confusion it caused.
> For the major concerns about our simulation results.  'In this setting, Model X should have an advantage, since it is specifically designed for Gaussian generated data.' I understand this sentence as describing the Fixed X Knockoff method in Barber 2014.  It is proposed for Gaussian outcomes with fixed X.
> 'Since the data were not Gaussian, the second-order matching method has the lowest power.' We wanted to express is that the X's in this setting are not generated from Gaussian distribution, but the model X (with second order matching) is proposed for X that generated from Gaussian distribution.
> 'The assumptions of the Fixed knockoff generations holds for the Gaussian cases, …" is to say the Fixed-X method is proposed for when the outcomes are generated from Gaussian. (linear regression models) Sorry for the confusion.
>
>  We will revise for a better  presentation of the paper.  And meanwhile, I would appreciate your letting us know if I have misunderstood your concerns and if there are other concerns you may have for our paper.
>
> Thanks.

---

> > ### Comment · AnonReviewer2 · 2018-11-06
> > **we need to be sure that such a Z_j exists for each X_j**
> >
> > No, this does not address the concern.  The paper you mention indeed points out that the VAE is aiming to find the Z that de-correlates X. But for the proof to hold, we need to be sure that such a Z_j exists for each X_j. This may not be the case in general.

---

> > > ### Author Response · Authors · 2018-11-19
> > > **Revision uploaded to address this question**
> > >
> > > Thanks much for the clarification.
> > >
> > > Please refer to section 3.1 in the revision, where we proved that for non-degenerate continuous and categorical $X$,   there always exist a latent $Z$, such that X |Z are mutually independent.
> > > And $Z$ also satisfies Var(X|Z) >0  for some Z with positive probability, so that if we generate knockoff \tilde{X} from X|Z,   $\tilde{X}$ and the observed $X$ are not identical.
> > >
> > > We also edited the simulation description and changed the label of the graphs to distinguish between the Gaussian/Non-Gaussian outcome with Gaussian/Non-Gaussian predictors.
> > >
> > > The theorem 3.3 (previous 2.2) although provide a bound for FDR, but it has limited implication for practice because the lack of general theoretical convergence rate for VAE.
> > > This is a common problem in all the emerging deep learning knockoff generators.
> > >
> > > To fill in the gap between current theory and practice, we provide more explanation and discussion regarding how to validate and evaluate the knockoffs generated from our framework in section 4, section 7, and appendix C.
> > >
> > > We look forward to hearing whether you would have a second opinion regarding our work based on these revisions to address your concern.
> > >
> > > Thanks

---

### Official Review · AnonReviewer3 · 2018-11-02
**The presentation of this paper can be improved. The notation is not very clear.**

**Rating:** 3
**Confidence:** 4

**Review:**

This paper proposes a new approach to construct model-X knockoffs based on VAE, which can be used for controlling the false discovery rate. Both numerical simulations and real-data experiments are provided to corroborate the proposed method.

Although the problem of generating knockoffs based on VAE is novel, the paper presentation is not easy to follow and the notation seems confusing. Moreover, the main idea of this paper seems not entirely novel. The proposed method is based on combining the analysis in ''Robust inference with knockoffs'' by Barber et. al. and the VAE.

Detailed comments:

1. The presentation of the main results is a bit short. Section 2, the proposed method, only takes 2 pages. It would be better to present the main results with more details.

2. The method works under the assumption that there exists a random variable $Z$ such that $X_j$'s are mutually independent conditioning on $Z$. Is this a strong assumption? It seems better to illustrate when this assumption holds and fails.

3. The notation of this paper seems confusing. For example, the authors did not introduce what $(X_j, X_{-j}, \tilde X_j, \tilde X_{-j} )$ means. Moreover, in Algorithm 1, what is $\hat \theta$ and $\hat f$.

4. I think there might be a typo in the proof of Theorem 2.1. In the main equation, why $\tilde Z$ and $\tilde X$ did not appear? They should show up somewhere in the probabilities.

5. In Theorem 2.2, how strong is the assumption that $\sup_{z,x} | log (density ratio)| $ is smaller than $\alpha_n$? Usually, we might only achieve nonparametric rate for estimating the likelihood ratios. But here you also take a supremum, which might sacrifice the rate. The paper suggested that $\alpha_n$ can be o( (n \log p)^{-1/2}). When can we achieve such a rate?

6. Novelty. Theorem 2.2 seems to be an application of the result in Barber et. al. Compared with that work, this paper seems to use VAE to construct the distribution $ P_{\tilde X| X}$ and its analysis seems hinges on the assumptions in Theorem 2.2 that might be stringent.

7. In Figure 1 and 2, what is the $x$-axis?

8. A typo: Page 2, last paragraph. "In this paper, we relaxes the ..."

---

> ### Author Response · Authors · 2018-11-19
> **Point-by-point response and clarification of the contribution**
>
> Thanks for the comments and here we provide point-by-point response to the questions and concerns.
>
>  ``This paper proposes a new approach to construct model-X knockoffs based on VAE, which can be used for controlling the false discovery rate. Both numerical simulations and real-data experiments are provided to corroborate the proposed method.   Although the problem of generating knockoffs based on VAE is novel, the paper presentation is not easy to follow and the notation seems confusing. Moreover, the main idea of this paper seems not entirely novel. The proposed method is based on combining the analysis in ''Robust inference with knockoffs'' by Barber et. al. and the VAE. ''
>
> Response:
> We summarized the contributions at the second paragraph of introduction section of the revision.
> Our main contribution is to provide a practical method to generate knockoffs. While writing and revising this paper, there are 2 other works on deep knockoff generators. One is based on GAN and the other is based on Generalized Moment Matching. We think the novelty of this work is comparable with these works and it remains for future investigation on systematic comparison of these methods.
>
> Detailed comments:
> 1. ``The presentation of the main results is a bit short. Section 2, the proposed method, only takes 2 pages. It would be better to present the main results with more details. ''
> Response:
> Thanks for the comment. For the readers to digest the idea of the paper, we have extended the presentation of the methods into 4 pages (excluding numerical results) in the revision. We added more motivation for the proposed framework and include a figure of Hidden Markov Random Field as a general example where a latent $Z$ can help generating $X$.  Section 3.1 to showed the existence of $Z$. Section 4 to explain the general strategy to evaluate generated knockoffs.
>
> 2. ``The method works under the assumption that there exists a random variable $Z$ such that $X_j$'s are mutually independent conditioning on $Z$. Is this a strong assumption? It seems better to illustrate when this assumption holds and fails.''
>
> Response:
> Thanks for the comment. We now added section 3.1 to discuss this in details.  In summation, the latent $Z$ always exist for non-degenerate $X$.  We illustrate a lot of examples as well provide Lemma 3.1 to justify the existence of Z.
>
> 3. ``The notation of this paper seems confusing. For example, the authors did not introduce what $(X_j, X_{-j}, \tilde X_j, \tilde X_{-j} )$ means. Moreover, in Algorithm 1, what is $\hat \theta$ and $\hat f$."
> Response:
> We now have added more explanation for the notations. We moved the notation explanation that appears inside Algorithm 1 to a separate paragraph: paragraph 4 section 3.2.
>
> 4. ``I think there might be a typo in the proof of Theorem 2.1. In the main equation, why $\tilde Z$ and $\tilde X$ did not appear? They should show up somewhere in the probabilities."
> Response:
> We did not find a typo in the proof of Theorem 2.1 (now 3.2). \tilde X does appear as \tilde X_j and \tilde X_{-j};  We considers \tilde Z  from distribution of Z|X, and argues that \tilde X generated from X|Z, with Z take the value of the generate \tilde Z, satisfies the knockoff's definition.
>
> 5.`` In Theorem 2.2, how strong is the assumption that $\sup_{z,x} | log (density ratio)| $ is smaller than $\alpha_n$? Usually, we might only achieve nonparametric rate for estimating the likelihood ratios. But here you also take a supremum, which might sacrifice the rate. The paper suggested that $\alpha_n$ can be o( (n \log p)^{-1/2}). When can we achieve such a rate?"
> Response:
> We clarified the limitation of Theorem 3.3 (previous 2.2) as a remark and moved the proof to the appendix. We also discussed further about the limitation of Theoretical guarantee for deep knockoff generators in section 7 and the alternative solutions.

---

> > ### Author Response · Authors · 2018-11-19
> > **Point-by-Point Responses-Continued**
> >
> > 6. ``Novelty. Theorem 2.2 seems to be an application of the result in Barber et. al. Compared with that work, this paper seems to use VAE to construct the distribution $ P_{\tilde X| X}$ and its analysis seems hinges on the assumptions in Theorem 2.2 that might be stringent."
> > Response:
> > Theorem 3.3 (previous 2.2) point out the link between our framework with the existing robustness analysis for knockoff generator.  But the contribution and novelty of this work is that it is one of the earliest work on proposing knockoff generators with deep learning models. We proposed a unique framework based on latent $Z$. It has the advantage of lighter computational cost because it focus on reconstruction of X from Z, instead of the more complex distributional discrepancy between X and \tilde X.
> > We demonstrate through simulation and real data application that this framework is a promising methods with reasonable FDR control in various settings.
> >
> >
> > 7. In Figure 1 and 2, what is the $x$-axis?
> > Response:
> > It is the signal magnitude. We describe it in the text. And now we added the explanation in the title of the figure.
> >
> > 8. A typo: Page 2, last paragraph. "In this paper, we relaxes the ..."
> > Thanks for the catch, we have revised and will continue to revise typos until the revision deadline.
> >
> > We look forward to hearing whether you would have a second opinion regarding our work based on these revisions to address your concern.
> >
> > Thanks

---

> > > ### Comment · AnonReviewer3 · 2018-11-28
> > > **Question about Theorem 3.2.**
> > >
> > > I appreciate the effort made by the authors to further improve this work. With respect to the revised version, I still have a question about Theorem 3.2.  It seems that in the first equation of its proof, you are conditioning on $\tilde Z$, so it should be the conditional probability given $\tilde Z$.
> > >
> > > In addition, I still reserve my concerns on Theorems 3.2 and 3.3.
> > >
> > > 1. It seems that the reason that $X$ and $\tilde X$ are knockoff pairs is because $X --> \tilde Z ---> \tilde X$ form a Markov chain. Is Theorem 3.2 itself a contribution of this paper?
> > >
> > > 2. In the proof of Theorem 3.2, it would be better if the authors make it clear what random variables are conditioning on. It seems that in the first equality, the authors are conditioning on $\tilde Z$, which yields $p_{X | Z} ( B_j, B_{-j} | \tilde Z) $. If this is true, then you should have $p_{Z | X} ( \tilde Z \given X) * p_X ( X \in A_j, A_{-j} )$.
> > >
> > > 3. For Theorem 3.3, it is still not very clear how strong the assumptions are. Is it possible to have an example showing how large $a_n$ can be?

---

> > > > ### Author Response · Authors · 2018-11-29
> > > > **Explanation of the proof for Theorem 3.2 and rate for Theorem 3.3**
> > > >
> > > > 1. Response for concerns with Theorem 3.2
> > > > Z is a latent variable that exists but not observed.
> > > > \tilde Z is the variable generated from the conditional distribution Z|X, thus (\tilde Z|X) and (Z|X) have the same distribution, i.e. p(\tilde Z|X)= P(Z|X). Therefore we have
> > > > 1). the joint distribution of (X, Z) and (X, \tilde Z) are also the same since p(X, Z)=p(X)p(Z|X).
> > > > 2). the marginal distribution of $\tilde Z$ and $Z$  are the same since p(Z)=\int _x p(x, Z) d F_X(x)
> > > > 3). The conditional distributions of  X|\tilde{Z} and X|Z is the same based on 1) and 2), since p(X|Z)=p(X, Z)/p(Z).
> > > >
> > > > In the current proof of Theorem 3.2, we skipped the above steps to justify that the right-hand side of the first equation in the proof is equal to that with all the Z's changed to \tilde Z.  Thanks for pointing that out, we would make Theorem 3.2 more easy to digest when we got the chance to submit the final revision.
> > > >
> > > > We think Theorem 3.2 itself is a contribution of this paper. It characterizes the key for the latent knockoff generation framework to work is the components of X need to be mutually independent given Z, i.e.  p(X_1,...X_p|Z)= p(X_1|Z)*p(X_2|Z)*...*p(X_p|Z).  This conditional independence is needed in proof for the equality between line 1 and line 2, where we factored P_{X|Z} into P_{X_j|Z} * P_{X_{-j}|Z}.
> > > >
> > > > We agree X->\tilde{Z}->\tilde{X} is a Markov chain. However, that is not a sufficient criterion for \tilde{X} is a knockoff of X.
> > > >
> > > >
> > > > 2. Response for a_n in Theorem 3.3
> > > > In the provided rate for Theorem 3.3, the asymptotic convergence of FDR needs a_n ~ o(n^{-1/2}).
> > > > For the nonparametric rate of a_n~ O(n^{-1/3}) or parametric rate a_n~ O(n^{-1/2}, they are too large for the bound of Theorem 3.3 to converge to q.
> > > > However, this does not mean our framework cannot control for FDR in practice.
> > > >
> > > > In practice, the distribution of X is usually estimated with a much larger sample size than the number of samples with the complete observation of both X and Y.  (refer to the original Model-X paper (Candes 2018)).
> > > > Denote N to be the number of data used to estimate the working models (for which we observed X, but may not have Y).
> > > > And denote n to be the sample size of complete pairs of (X,Y), which is used for variable selection.
> > > > Assume a_n ~ O(N^{-b}), with some b>0.
> > > >
> > > > Then we just need O(N^{-b}) ~ o(n^{-1/2}) to guarantee FDR bound converge to q. This is not hard to achieve when we can collect X more easily than collecting Y.
> > > >
> > > > Thanks for pointing out. We would revise Theorem 3.3 and its remark to reflect this point when we can submit another revision.
> > > >
> > > > We hope these responses resolve the concerns and let us know if further explanation is needed.
> > > >
> > > > Thanks much.

---

### Author Response · Authors · 2018-11-25
**Summary of Revision**

Dear Reviewers and Chairs,

Thanks for the helpful feedback, we have made the following revision to address the comments and concerns.
The highlight of the contribution of our paper is that it is among the first papers to tackle the problem of knockoff generation with deep learning,  which is an emerging important application of representative learning.  We offers a promising general framework through the latent variable $Z$. The pros and cons compared to the other alternative deep learning frameworks are waiting to be further investigated.

1. We added section 3.1 to provide theoretical justification for the existence of latent variable $Z$ for any $X$ from non-degenerate distributions (continuous or categorical).  For the continuous $X$, we provide Lemma 1 and its proof in section 3.1.
We also provide examples and a new graph Figure 1 to illustrate the relationship the latent Z and X in Hidden Markov Random Field .  We also added more conceptual explanation for our framework in section 3.2.

2. We split the original introduction section into two sections, add more conceptual explanation of the Model-X knockoff, and provide summary of contributions in paragraph 2.

3. We moved the proof of theorem 3.3 (previous theorem 2.2) to appendix A. We added a remark after the theorem 3.3.  In the remark we clarifies the limitation of the current theory in FDR guarantee, when the rate a_n in theory 3.3 is not guaranteed.
(See our further discussion with Reviewer 3 in the forum that even a_n rate is larger than n^{-1/2}, with a larger training sample size of X than the sample size of (X,Y), our method still can achieve FDR guarantee.)

4. We added a section 4 Goodness of Fit to give a comprehensive description for how to validate deep learning generated knockoffs in practice.  We also propose a way to check the model fitting for our framework is to check X-\hat{f}(\tilde{Z}) is mutually independent. Some preliminary results of graphical check of this goodness of fit diagnostic is included in appendix C.

5. We moved the implementation details of the simulations to appendix B. We changed the titles the simulation figures for clarity.

6. We revised the discussion section. We moved the previous discussion about independence check to section 4 goodness of fit. We briefly discussed about how our current implementations performed with the graphical check. We added some discussion about the emerging other framework of Deep knockoff Generator in discussion section and further clarifies the unique contribution of our framework to this emerging subfield of representative learning.

We would like to mention that there is a submission of another paper on using deep learning to generate Knockoff for ICLR 2019, entitled 'KnockoffGAN: Generating Knockoffs for Feature Selection using Generative Adversarial Networks'.  The two papers although used different deep generative models, but are comparable in contribution, experiment results and judging the interests of ICLR attenders in this new topic.

We hope these revisions would resolve the previous concerns and facilitate a favorable decision of our work.

Thanks

---

### Meta-Review · Area_Chair1 · 2018-12-14

**Confidence:** 3
**Recommendation:** Reject

**Metareview:**

The paper presents a novel strategy for statistically motivated feature selection i.e. aimed at controlling the false discovery rate. This is achieved by extending knockoffs to complex predictive models and complex distributions; specifically using a variational auto-encoder to generate conditionally independent data samples with the same joint distribution.

The reviewers and ACs noted weakness in the original submission related to the clarity of the presentation, relationship to already published work, and concerns about the correctness of some main claims (this mostly seems to have been fixed after the rebuttal period). There are additional concerns about a thorough evaluation of the claimed results, as the ground truth is unknown. The authors (and reviewers) also note a similar paper submitted to ICLR with the same goal but implemented using GANs. Nevertheless, there remain significant concerns about the clarity of the presentation.